

# Pesticide transport through the vadose zone under sugarcane in the Wet Tropics, Australia

Rezaul Karim[1,2], Lucy Reading[1], Les Dawes[3], Ofer Dahan[4], Glynis Orr[5]

[1]School of Biology and Environmental Science, Queensland University of Technology, Brisbane, 4000, Australia.
[2]Department of Environmental Science and Technology, Jashore University of Science and Technology, Jashore, 7408, Bangladesh.
[3]School of Civil and Environmental Engineering, Queensland University of Technology, Brisbane, 4000, Australia.
[4]Department of Environmental Hydrology and Microbiology, Zuckerberg Institute for Water Research, Ben-Gurion University of the Negev, Sede Boqer Campus, 84990, Israel.
[5]Department of Environment and Science, Queensland Government, Cairns, 4870, Australia.

*Correspondence to*: Lucy Reading (lucy.reading@qut.edu.au)

**Abstract.** Photosystem II (PS II) pesticides, recognised as a threat to ecological health, were targeted for reduction in sugarcane farming in the Great Barrier Reef (GBR) catchments. Alternative herbicides, the non-PS II herbicides (including glyphosate, paraquat, 2,4-D, imazapic, isoxaflutole, metolachlor, and S-metolachlor), continue to be used in these
catchments. However, the potential ecological fate, transport, and off-site environmental effects of non-PS II herbicides, with respect to their usage scheme, local rainfall patterns, and infiltration dynamics, has not been investigated previously. A vadose zone monitoring system, instrumented beneath a sugarcane land in a GBR catchment, was applied for real time tracing of pesticide migration across the unsaturated zone, past the root zone during 2017-2019.The monitoring of regularly applied pesticides (fluroxypyr and isoxaflutole), exhibited substantial migration through the unsaturated zone. Within one
month after application of fluroxypyr, it leached to 2.87 m depth in the vadose zone, with declining concentrations with depth. Isoxaflutole, which was applied yearly, was found only once, in November 2018, at 3.28 m depth in the soil profile. Other pesticides (imazapic, metolachlor, glyphosate and haloxyfop), applied at the same period, were not detected through the vadose zone. However, imidacloprid, which was not applied at the site during the monitored period, was detected across the entire vadose zone, revealing substantial resistance to degradation. The results show no evidence of any regularly applied
pesticides in the site bores at the end of the study, indicating their ultimate degradation within the vadose zone before reaching the groundwater.

## 1 Introduction

Agriculture has a considerable influence on groundwater quality as well as quantity. With continuous increases in agricultural production, agrochemicals (fertilizers and pesticides) can be transported into the environment (Adeoye et al.,
2013). These agrochemicals can potentially percolate through the rhizosphere after rain and may ultimately leach to the underlying groundwater, potentially lowering the quality of the groundwater (Adeoye et al., 2013). During the past centuries,



the catchments close to the Great Barrier Reef (GBR) have been considerably changed, through the growth of agricultural activities and municipal settlement in these catchments. There is evidence that this development is resultant in water quality deterioration in the GBR waterways (Brodie et al., 2013; Armour et al., 2009; Rasiah and Armour, 2001). Despite the high

level of protection in recent decades, the situation of water quality has continued to decline (Kroon et al., 2016). The key concerns for the catchments are the increasing quantity of suspended sediments, the discharge of nitrogen (Brodie et al., 2015) and the transport and potential toxic effects of pesticides from the farming areas (Smith et al., 2012). Many studies have detected increased loads of agrochemicals e.g., fertilizers and pesticides being transported to the GBR via runoff (Smith *et al.*, 2012). Yet, there is also potential for pesticides and nutrients to be moved to the GBR via groundwater and only a few

studies have focused on groundwater pathways (Rayment, 2003; Stieglitz, 2005; Armour et al., 2009; Rasiah and Armour, 2001).

In 2001, the Great Barrier Reef Marine Park Authority first reported the water condition deterioration in the GBR (GBRMPA, 2009). In response, the Reef Water Quality Protection Plans 2003, 2009 and 2013 and lastly, the Reef 2050 Plan were designed to deal with the problem (Brodie et al., 2017). During this period 2001-2018, there has been a significant

growth in the knowledge of pesticide dynamics including sources, movement, exposure and fate, and finally environmental threat of pesticides to the GBR (Devlin et al., 2015; Johnson and Ebert, 2000; Kroon et al., 2013; Shaw et al., 2010; Smith et al., 2012). Pesticides in the GBR area have been discovered in rivers and creeks (Smith et al., 2012); sediments (Haynes et al., 2000); freshwater wetlands (Devlin et al., 2015; Davis et al., 2012) and marine environments (Gallen et al., 2016; Shaw et al., 2010). Elevated pesticide concentrations were mostly found in connection with sugarcane farming in three

geographical regions the Tully–Murray, Burdekin–Townsville and Mackay Whitsunday region of the GBR catchment area (Lewis et al., 2009). Pesticides were also observed in the Johnstone River (Wallace et al., 2015) and South Johnstone River (Smith et al., 2012), which are also sugarcane dominated catchments.

Five photosystem (PS) II restraining herbicides (namely tebuthiuron, ametryn, hexazinone, diuron, and atrazine) were recognised as a threat to the environmental health and resilience at the GBR catchments and targeted decrease in land

management practices (Brodie et al., 2009; Davis and Pradolin, 2016; Devlin et al., 2015; Masters et al., 2013; Silburn et al., 2013; Thorburn et al., 2013; Vardy et al., 2015; Wallace et al., 2015). The pesticide reduction loads, reducing 50% of end-of-catchment loads, were introduced in agriculture primarily from cropping and sugarcane farming through the best management practices (Devlin et al., 2015). Reported management practices already in use across sugarcane farming not only in the GBR but also globally, include cropping system (Nachimuthu et al., 2016); selection of product, timing of

application, precision application (García-Santos et al., 2016; Melland et al., 2016; Masters et al., 2013; Oliver et al., 2014), using a straw/trash blanket (Dang et al., 2016; Nachimuthu et al., 2016) or even switching to 'alternative PS II or non-PS II herbicides'(Tao and Yang, 2011; Lewis et al., 2016). The alternative/other non-PS II herbicides including 2,4-D, glyphosate, paraquat, monosodium methyarsenate (MSMA), MCPA, imazapic, trifloxysulfuron sodium, isoxaflutole, trifluralin, S-metolachlor, metolachlor, and pendimethalin, have been widely used in sugarcane farming (Davis et al., 2014).



At the paddock scale, pesticide loads, estimation of the mass of pesticides annually entering to the GBR from agriculture, were quantified based on monitoring data and modelling data (Australian Government and Queensland Government, 2018). Significant advances in techniques to measure impacts in situ were applied to determine the default values through the species sensitivity distribution technique (Smith, 2018). The aquatic ecosystem protection guideline values have been developed recently for 28 pesticides, including 13 PS II herbicides, frequently utilized in the GBR catchments (King et al.,

2017b, a; Smith, 2018). However, out of all pesticides detected, PS II herbicides still provide the highest loads and the highest ecological risk to the GBR (Devlin et al., 2015; Wallace et al., 2015; Gallen et al., 2016; Vardy et al., 2015). These potential risks have been supported by the detection of several PS II herbicides in soil, vadose water, and groundwater at low concentrations (Karim et al., 2021).

The vadose zone covering from the ground surface to the aquifer, is incredibly complicated in structure, governing water

passage from the land surface to aquifers (Arora and Ahmed, 2011). Dahan et al. (2009) introduced the vadose zone monitoring system (VMS). The system consists of flexible TDR waveguides (FTDR) and flexible vadose zone sampling ports (VSP) which support attachment of the water content sensor and sampling units to the unusual structure of borehole walls. The VMS enables uninterrupted tracing of water percolation and chemical transport across the entire vadose zone (Rimon et al., 2007; Dahan et al., 2009; Turkeltaub et al., 2016; Aharoni et al., 2017). The installation of VMS has been

tested internationally in various hydrological setups. Examples include those in largely desert and semi-arid areas in Rene, California (Dahan et al., 2003), in the hyper-arid desert, the Arava Valley, Israel (Dahan et al., 2007), in a typical desert ephemeral river, Namibia (Dahan et al., 2008), and also in the Mediterranean climate, Israel (Rimon et al., 2007). It was also applied for PS II herbicides transport across the vadose zone in the sugarcane field during 2017-2019 (Karim et al., 2021). However, this study was unable to link the water percolation and pesticide migration dynamics with application regime, as

PS II herbicides (including Priority Five) were then not applied during monitoring period at the site.

It is important to track pesticide transport and transformation dynamics from shallow to deep vadose zone horizons, in order to achieve 'real time early warning' on pesticide pollution potential (Dahan et al., 2009) and thereby quickly detect and take steps for managing those pesticides (Lewis et al., 2016). The lag times between application to appearance of pesticides in groundwater or impact on environment may range from days to several years (Bidwell, 2000). Therefore, continuous

tracking of the transport characteristics of pesticides (after immediate application) in the unsaturated zone is critical for appropriate assessment of the possible impact of pesticide usage on groundwater. The transport and transformation of pesticides in the unsaturated zone is not yet clearly understood due to complex biotransformation process that can take place in the unsaturated zone (Rivett et al., 2011). There is scarce information on the possible ecological fate and off-site environmental impacts of non-PS herbicides used in sugarcane farming catchments (Davis et al., 2014).

This study aims to investigate water percolation and pesticide migration dynamics (with application regime) across the vadose zone below sugarcane fields in the Wet Tropical conditions in Australia. The study was performed using continuous measurements of temporal variations in soil saturation plus measurements of the concentration of pesticides along the vadose zone profile and underlying alluvial aquifers at sugarcane fields at South Johnstone River sub-basin, Australia. A vadose



zone monitoring system was set up to enable the characterization of pesticide (non-PS II herbicides) migration with respect
to pesticide application, sugarcane growing period and finally, rainwater infiltration.

## 2 Methods and Material

### 2.1 Monitoring site details

The study area (17°44'44.72" S 146° 2'58.76" E; Fig. 1) is in sugarcane field near Silkwood township in the centre of the
South Johnstone catchment. It is owned by a commercial sugar manufacturing company, MSF, and is approximately 2.8 ha.
It is isolated from the surrounding surface water run-on by a constructed channel. The channel (1–1.5 m deep) bounds the
northern, eastern, and southern borders of the paddock, enabling drainage away from the site (Masters et al., 2017). The
construction of channel around three sides of the paddock is not typical practice in the catchment.

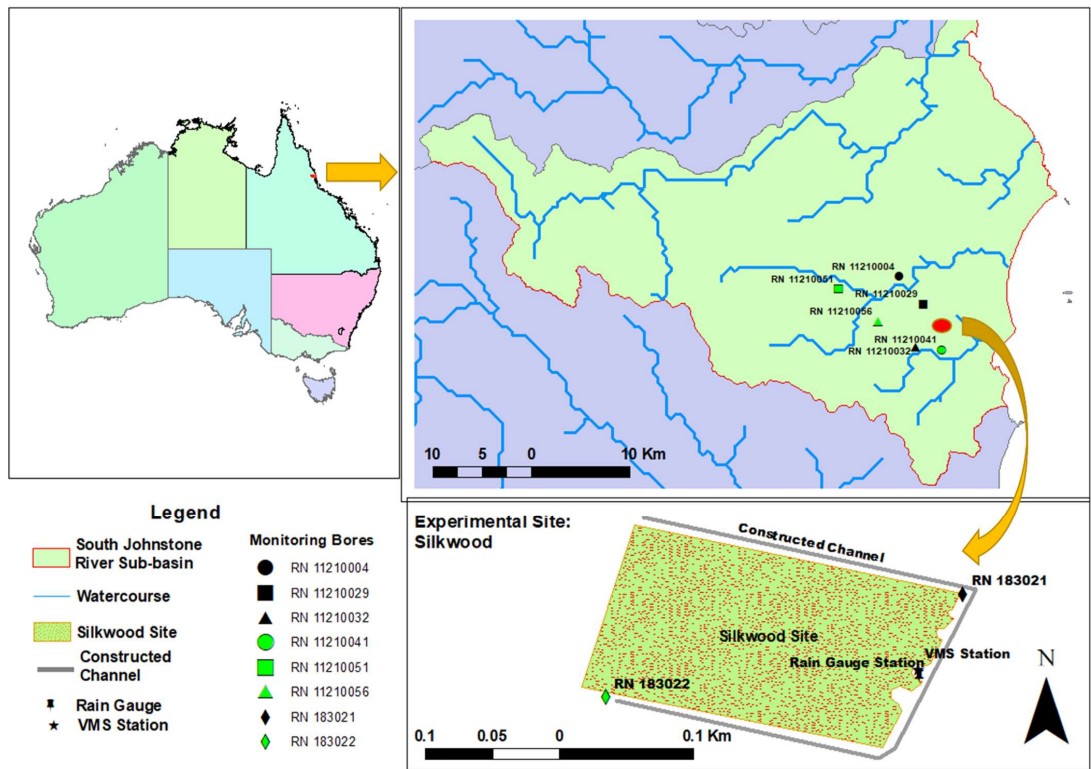

**Figure 1.** Monitoring site: Silkwood, South Johnstone Sub-basin, Queensland, Australia.




A detailed soil survey was carried out across the monitored site to identify and map the soil types (Masters et al., 2017). The dominant soil is regionally described as Bulgun series, a poorly drained alluvial soil first described by Murtha (1986) and also classified as a very deep dermosolic, redoxic, humose-acidic, dermosolic, redoxic, hydrosol (Isbell, 2016). The detailed soil profile, to 4.00 m depth, was stratified into five different layers, predominantly loamy textures (Fig. 2) (Karim et al., 2021). The extended profile to 12.00 m depth was predominantly composed of clay rich layers, followed by aquifer material

at 9.00 – 11.00 m depth (Stanley, 2019). These loam and clay rich textures regulate the site hydrology (infiltration, runoff, and deep drainage). The organic matter content of productive agricultural topsoil is usually between 3 and 6% (Fenton et al., 2008) but the present study found higher organic matter, above 20% in the topsoil (0.0-0.7 m depth, Table 1), than the previous study (5.7%, 0.0-0.1 m depth) in wet conditions (Masters et al., 2017).

**Table 1.** Soil organic matter at the site soil (Loss-On-Ignition, LOI, Method).


| Depth (m) | 0-0.15 | 0.15-0.40 | 0.40-0.70 | 0.70-1.25 | 1.25-2.20 | 2.20-2.90 | 2.90-3.65 | 3.65-3.90 |
|---|---|---|---|---|---|---|---|---|
| Organic matter (%) | 20.8 | 23.07 | 17.26 | 3.36 | 14.29 | 9.58 | 6.01 | 8.38 |

The study site experiences a humid tropical climate, predominantly influenced by coastal meteorological situations (Tahir et al., 2019). The annual average rainfall is 3,092 mm/yr at the nearest Australian Bureau of Meteorology (BoM) Station–

Bingil Bay (Site 032009), situated nearly 12.5 km of the site. This area typically experiences the highest rainfall in the wet seasons during December -April). For regular rainfall monitoring, the site was fitted with tipping bucket rain gauges, which were organized by the Queensland Government (Masters et al., 2017). The site is not irrigated, which is common in the tropical conditions for sugarcane cultivation. Therefore, the local rainfall patterns govern the site hydrology with respect to soil stratigraphy.


**2.2 Cropping history of the monitoring site**

After harvesting the previous sugarcane crop in November 2013, the site was laser-levelled to ensure a uniform slope across the paddock. Then, planting beds were designed using controlled traffic farming methods in early December (Masters et al., 2017). In the subsequent wet season 2013-14, the land was uncultivated. After the wet season, lime was applied. The cane

variety, Q183 was transplanted in July 2014. Typically, sugarcane becomes mature in spring (September- November) and is harvested at the end of this season in the Wet Tropics regions. In this article, the period from September to October will be considered as a sugarcane growing season (plant or ratoon, the new shoot springing from the base). The cropping periods will be referred to as 2014–15 (plant), 2015–16 (ratoon 1), 2016–17 (ratoon 2), 2017–18 (ratoon 3), 2018–19 (last harvesting, ratoon 4) and 2019-2020 (the fallow, field processing for the next plantation) (Table 2).




**Table 2.** Pesticide applications during 2013-2019.

| Pesticides | Units | 2013-14 | July 2014 | Nov 2014 | 11 Nov 2015 | Nov 2015 | 25 Nov 2016 | Dec 2016 | 2 Nov 2017 | 18 Dec 2017 | 4 Oct 2018 | 10 Oct 2018 | 9 Oct 2019 | 21 Nov 2019 | July 2020 |
|---|---|---|---|---|---|---|---|---|---|---|---|---|---|---|---|
| | | Field Preparation | | | Plant | Ratoon 1 | | Ratoon 2 | | Ratoon 3 | | Ratoon 4 | | | Fallow | |
| Imidacloprid | g/ha | | | | | | | | | | | | | | |
| Imazapic | g/ha | | | 75 | | | | | | | | | | | |
| Metolachlor | kg/ha | | | 2 | | | | | | | | | | | |
| Fluroxypyr | g/ha | | | | | | | | 100 | | | | | | |
| Isoxaflutole | g/ha | | | | | 150 | | 150 | | 150 | | 150 | | | |
| Glyphosate | L/ha | | 12 | | | | | | | | | | | 4 | |
| Haloxyfop | L/ha | | | | | | | | | | | | | 0.3 | |
| (status) | | Bare field | Plantation | | Plant was harvested. | | 1st Ratoon was harvested. | | 2nd Ratoon was harvested. | | 3rd Ratoon was harvested. | | 4th Ratoon (last) was harvested. | | Fallow for the next plantation |

A succinct description of all cultivation and pest and weed management methods during the planting to harvesting is given in
the Table 2. Each year, after harvesting the matured ratoon, pesticides were applied for the next ratoon at the site. Pesticide
applications at the site included: imazapic and metolachlor in November 2014, fluroxypyr in November 2017, and
isoxaflutole each year after harvesting. Glyphosate was applied during 2013-2014 and 2019-2020 and haloxyfop applied
once in November 2019.

### 2.3 Monitoring setup

### 2.3.1 Vadose zone monitoring system

The site was instrumented with a VMS (by Sensoil Ltd, Israel) in July 2017 (Fig. 2). The detail of the VMS, installation
procedures and performance have been described previously (Rimon et al., 2007). There are two sleeves (A & B) in the
installed VMS at the site. Each sleeve is installed in a slanted (35° to the perpendicular) borehole. Each sleeve consists of
four FTDR for constant measurements of soil water content and four VSPs for regular porewater sampling (Fig. 2) (Karim et
al., 2021). The slanted holes were bored using a sonic drilling rig.



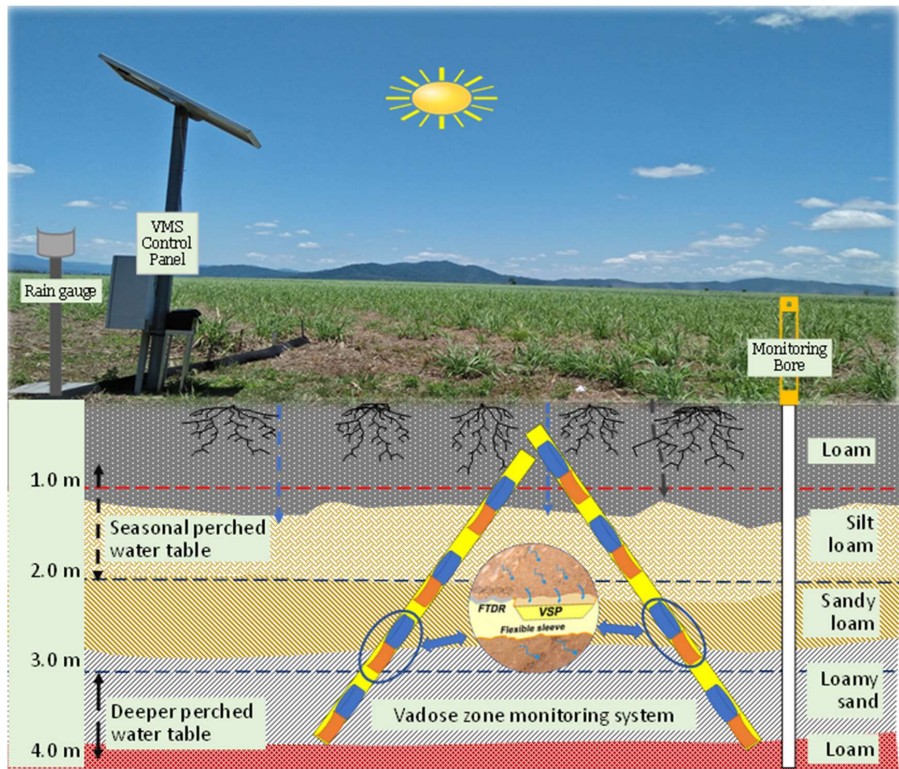

**Figure 2.** Schematic of the VMS installed in the monitoring site (July 2017).

The vertical allocation of VMS units (FTDR and VSP) along with soil profile (texture and clay content) is given in Fig. 3. While sampling, pressurised gas ($N_2$) is utilized to recover porewater to the sampling port (Dahan et al., 2009). The site

VMS system was regulated with data acquisition and logging devices (e.g., CR800) (Karim et al., 2021).



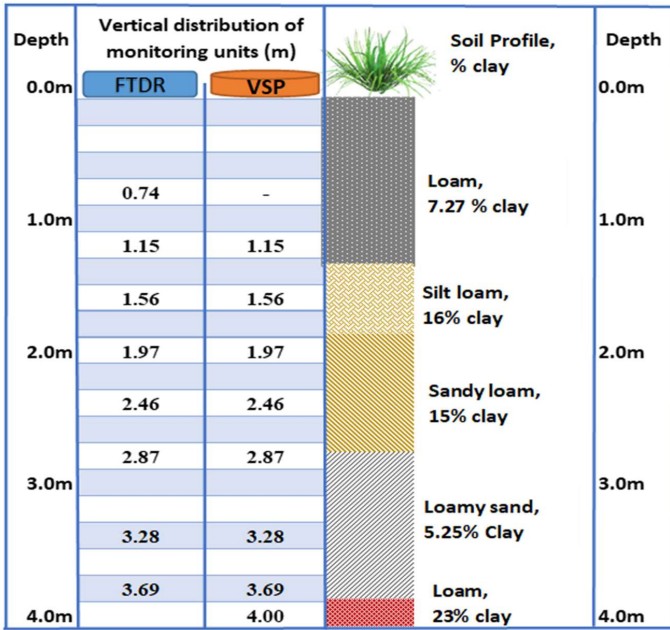

**Figure 3.** Vertical distribution of the VMS units and soil profile with clay content.

### 2.3.2 Monitoring bores

Six state government listed groundwater bores namely RN (registered number) 11210004, RN11210029, RN11210032,
RN11210041, RN11210051 and RN11210056, were selected in the South Johnstone River sub-basin. The aquifer depth of
these bores ranged from 7 to 33 m (Stanley, 2019). Additionally, two bores, RN 183021, and RN 183022 (aquifer depth, 9-
11 m) were installed in the northeast and southwest corner of the paddock, respectively. The surface elevation of the
installed two bores (RN 183021 & 183022) was 7-7.5 and 5-5.5 m, respectively, indicating the surface water flow from
southwest to northeast directions (Masters et al., 2017). These bores were fitted with pressure transducers to monitor the
170    ground water level (GWL) at the site. A perched aquifer at 3-4 m depth was encountered, while drilling bore RN 183021.

### 2.4 Sampling

Water and soil sampling at the site were scheduled from July 2017 to December 2019, considering the sugarcane growing
phase and the pesticide application time. The half-lives of pesticides applied recently at the site (Table A1) were also
considered for the sampling plan.



### 2.4.1 Water sampling

During the monitoring period, water samples (a total of 56 samples) were gathered from designated monitoring bores and from eight VSPs at different depths across the vadose zone (Table 3). For ground water sampling, each bore was expunged of 3 x volume to eliminate static bore water before sampling and confirm a more precise representation of aquifer water (Sundaram et al., 2009). Samples were stored in cool conditions at the site and kept in storage at 4°C during transport to the lab for chemical analysis.

**Table 03.** Water sampling regime for pesticide analysis (√ denotes sampling regime).

| Sampling date | Groundwater at the regional aquifer (six bores) | Vadose water through the VSP and groundwater from two bores at the site |
|---|---|---|
| 13, July 2017 | √ | X |
| 02, November 2017 | X | √ |
| 11, January 2018 | X | √ |
| 9, May 2018 | X | √ |
| 15, November 2018 | X | √ |
| 1, December 2019 | X | √ |

### 2.4.2 Soil sampling

Based on pesticide application regime (before and after), soil samples (a total of 12 samples) were collected in 2018 and 2019 (Table 4). Soil sampling cores were collected from a depth of 20 cm from the soil surface at intervals to 55 cm by hand driven auger and Geo Probe method. These were immediately cooled on ice and kept cool during transport to the laboratory.

**Table 4.** Soil sampling schedule for pesticide analysis (√ denotes sampling regime).

| Sampling method | Sampling date | Sampling depth | | |
|---|---|---|---|---|
| | | 25 cm | 40 cm | 55 cm |
| Hand auger | 18, September 2018 | √ | √ | √ |
| Hand auger | 11, November 2018 | √ | √ | √ |
| Geo Probe method | 11, November 2019 | √ | √ | √ |
| Hand auger | 3, December 2019 | √ | √ | √ |



**2.5 Chemical analysis**

The pesticides and their possible metabolites were analysed by Queensland Health, Coopers Plains, Queensland. All collected samples (water and soil) were separated via solid phase extraction (SPE) and examined by liquid chromatography mass spectrometry (LCMS). The analytical methodology (SPE combined with LCMS) is generally utilised by the Great Barrier Reef Catchment Loads Modelling Program (Gallen et al., 2016; Shaw et al., 2010; Vardy et al., 2015; Wallace et al., 2015). One insecticide (imidacloprid) and six herbicides namely imazapic, metolachlor, fluroxypyr, isoxaflutole, glyphosate and haloxyfop, which have been applied in recent years at the monitoring site, were tested in this study (Table 5 and Table A1).

**Table 5.** Ecotoxicity threshold value of Pesticides and their common metabolites.

| Class | Pesticides (Non-PS II) | Detection limit, (µg/L) [a] | Ecotoxicity threshold value [b],* (µg/L) | Common metabolites [d] |
|---|---|---|---|---|
| N-acetylcholine receptor modulators | Imidacloprid | 0.02 | 0.11 | |
| Amino acid inhibitors | Imazapic | 0.01 | 0.41 | |
| Long chain fatty acid inhibitors | Metolachlor | 0.005 | 0.71 | |
| Auxin growth regulators | Fluroxypyr | 0.005 | 160 [c] | 4-amino-3,5-dichloro-6fluoropyridin-2-ol, |
| | | | | 4-amino-3,5-dichloro-6-fluoro-2-methoxypyridine |
| Meristematic tissues growth inhibitors | Isoxaflutole | 0.02 | 0.46 | 2-cyano-3-cyclopropyl-1-(2-methylsulfonyl-4-tri-uoromethylphenyl) propan-1,3-dione) (DKN) |
| EPSO enzyme synthesis inhibitors | Glyphosate | 0.7 | 250 | Alpha-amino-3-hydroxy-5-methyl-4-isoxazole propionic acid (AMPA) |
| Acetyl-coenzyme A carboxylase | Haloxyfop | 0.02 | 2000 | |

[a] The detection limits were defined by the Analysis suite for water samples, developed by QHFSS.

[b] King et al. (2017b), [c] King et al. (2017a), [d] PPDB(2021)

* Ecotoxicity threshold value were derived using the Burrlioz 2.0 (2016) software for 95% species protection (Warne et al., 2015)



## 3 Results and discussion

Increases in sediment water content are typically the outcome of water infiltration and downwards propagation of a wetting front, while decrease of water content is a consequence of either deep drainage or evapotranspiration at the shallow layers affected by the root uptake (Dahan et al., 2008). Accordingly, sequential wetting with depth represents the wetting front propagation velocity and infiltration fluxes (Rimon et al., 2011). Firstly, the site hydrology (Section 3.1) with respect to local rainfall pattern, site morphology and sugarcane growing season will be defined. Then, pesticide migration will be characterised with respect to the site hydrology (Section 3.2). Finally, the pesticide concentrations in groundwater will be reported and compared with the ecotoxicity threshold value (Section 3.3).

### 3.1 Site Hydrology (Infiltration, and GWL variations)

### 3.1.1 Rainfall infiltration

Temporal changes in sediment water content through the vadose zone signifies percolation events which are well corelated to rainfall intensity and frequency (Fig. 4). Substantial increases in the measured water content of shallow sediments (0.74 m and 1.15 m) were observed immediately following all significant rainfall events. However, the next two deeper probes, at 1.56 m and 1.97 m, show high (saturated) water contents that are caused by a seasonal perched water table. Indications for seasonal perched water table were obtained during the drilling for VMS installation. Deeper in the cross sections at depths of 2.46 m and 2.87 m, under the uppermost perching layer, the water content sensors exhibited unsaturated conditions with small fluctuations that are attributed to the rainfall events. Finally, the deepest probes (3.28 m and 3.69 m), exhibit high and steady water content values that represent saturation over the whole monitored period due to a perched layer at this depth.

Each significant rainfall event initiated the infiltration process by causing substantial increases in water contents, notably at shallow sediments (0.74 m and 1.15 m). The 158.4 mm rainfall event on 19 September 2017 resulted in the first sharp rise in water content, indicating the appearance of wetting front at the first probe at 0.74 m below the ground surface. This wetting front was marked as an increase in moisture content from 20% to 46% at 0.74 m (A). After this response, this probe experienced a reduction in the water content. This could have been caused by drainage, evaporation and/or water uptake by mature sugarcane (ratoon 2). Sequentially, an increase from 26% to 42% was recorded at 1.15 m depth (within three hours), indicating that the wetting front propagated to that depth.



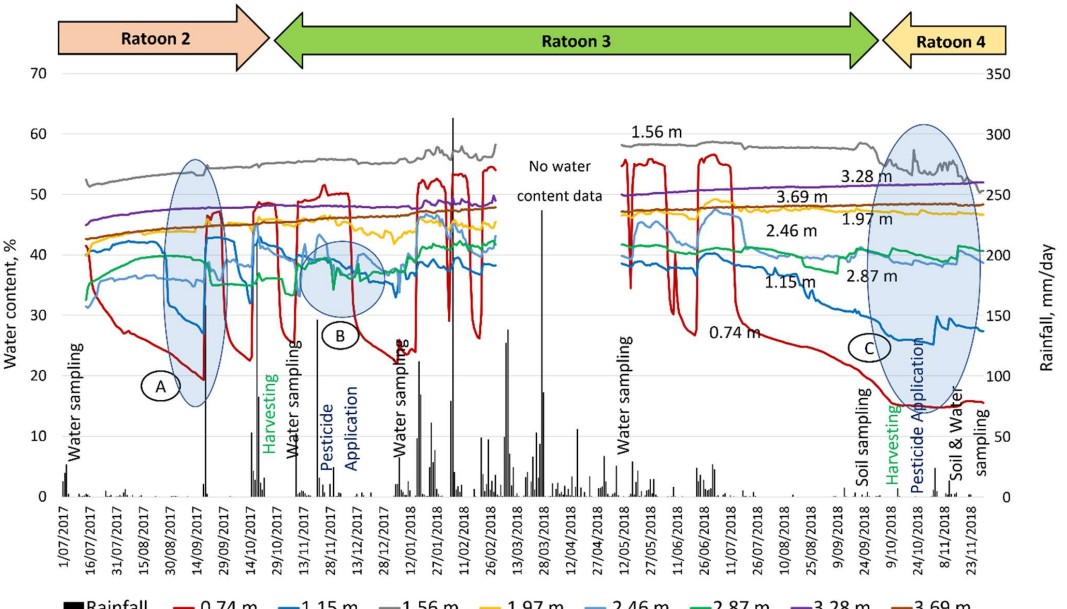

**Figure 4.** Variations in water content in the vadose zone along with rainfall.

The next two probes at 1.56 m and 1.97 m, exhibited high and stable water content (>50%), with no significant change over

time, indicating stable soil saturation. This could be due to the existence of a seasonal perched water table around 1-2 m depth in the vadose zone (Masters et al., 2017). The responses of these probes also relate to this layer's composition, fine sediment (clay and silt, 76%).

The probes at 2.46 m and 2.87 m, situated in the sandy loam layer, showed lower water content than the upper probes (at 1.56 m and 1.97 m), but showed large fluctuations over time, indicating unsaturated conditions under the seasonal perched

water table (1-2 m depth). As the sandy loam layer has low water retention properties, it can quickly drain to the deeper layers, which was evidenced by a reduction water content over time (B).

As a result of the continuous deep drainage from the unsaturated layers, water contents were consistently high (~50%) at the deepest two probes (3.28 m and 3.69 m), suggesting their permanent saturation. The underlying loam layer (with 23% clay content at 3.80-4.00 m depth (Fig. 2) serves as hydrological barrier, reducing further downward water percolation. Therefore,

the two probes at 3.28 m and 3.69 m recorded high-water contents, implying their location within a deeper perched layer. Indications for a perched water table (3-4 m depth) were obtained, while drilling the bore of RN 183021 at the monitoring site, near the site VMS station.





During the dry period (July-November 2018), all eight FTDR probes across the vadose zone (C) gradually responded with a declining trend in water content. The upper two probes reduced in water content from saturation to 15% at 0.74 m and 25% at 1.15 m depth. During the dry period, mature sugarcane can access a significant amount of water from the seasonal water table (within 1.5 m depth) (Hurst et al., 2004), significantly reducing the water content in the upper soil layers. Following this recession at 0.74 m and 1.15 m, the water content at probe at 1.56 m depth dropped from saturation to ~50% for the first time. The declines in water content in the upper layers also aided to a gradual rise (~2-3%) in the water content at the deepest two probes (3.28 m and 3.69 m).

### 3.1.2 Ground water level (GWL)

Both bores (RN 183021 and 183022) showed a substantial fluctuation in ground water level (GWL) in response to the rainfall at the site during November 2017- December 2019 (Fig. 5). The GWL trends showed shallower depths for bore 183021 (<2 m depth and even above ground) and 2 to 3.80 m depth from the surface for bore 183022 during the monitoring period. Yet, the aquifer material (medium to coarse sand layer) of both bores is encountered at the depth of 9-11 m. The shallower GWL depth (< 4 m or even above the ground) could be due to the semi-confined nature of the aquifer at 9-11 m and regular degree of saturation of the dense clay rich layers at the site (Fig. 3).

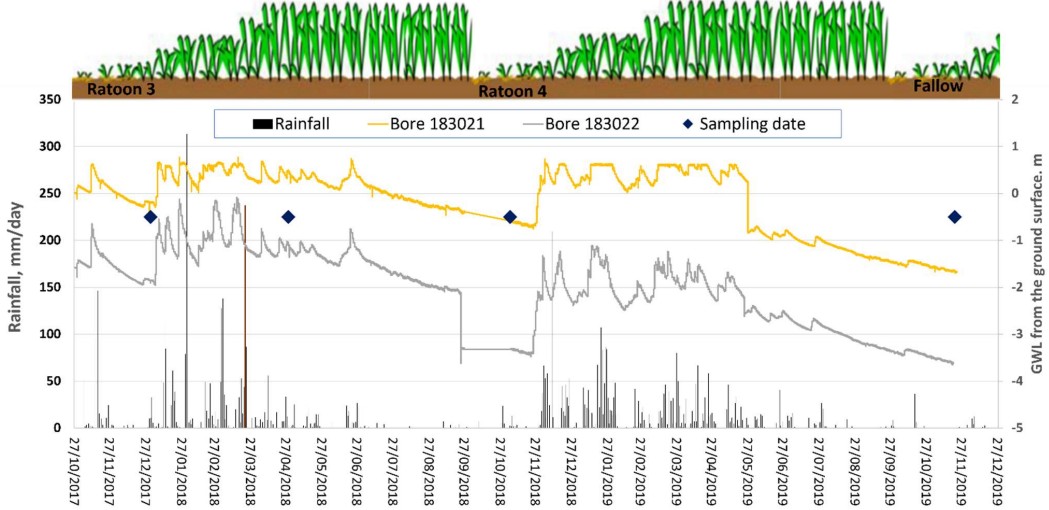

**Figure 5.** Ground water level (GWL) of Bore 183021 and 183022 with respect to local rainfall (GWL was measured from the ground surface).

There was also a discrete difference in the GWL trends between bore 183021 and bore 183022. They are approximately 265m apart but showed with a 2 m difference in GWL. The previous study by Masters et al. (2017) showed ~1-2 m elevation difference based on elevation model (m) at the site. This topographic deviation could be the reason for the deviation in GWL




trends. There was a distinct variation in GWL trends between wet/rainy seasons (December – April) and dry period (September – November). At the beginning of Ratoon 3, the GWL was close to surface for bore 183021 and within 2 m

depth for bore 183022. Quick responses in GWL's were observed with a significant rainfall event, for example a light rainfall event 52.2 mm, 9 Nov, and moderate rain event 146 mm/day on 22 Nov 2017, resulted in peaked GWL's for both bores (Fig. 5). While flooding was observed at the monitoring site in March 2018, groundwater was observed to overflow, at the top of the bore casing, for 183021 and become close to the surface for 183022. The groundwater responses at this site support the infiltration data (Fig. 4) which shows the movement of wetting fronts throughout the entire vadose zone, passing

through perching layers. The rapid responses observed for groundwater at this site provide evidence of the close connection between the surface and underlying groundwater.

**3.2 Pesticide transport through the vadose zone**

In the present study, during 2017-19, seven commonly used non-PS II herbicides namely imidacloprid, imazapic, metolachlor, fluroxypyr, isoxaflutole, glyphosate and haloxyfop (Table 5), were analysed in the soil samples (to 50 cm) and

in vadose water sampled through the vadose zone (1.15 to 4.00 m depth). Fluroxypyr (half-life, 51 days) and isoxaflutole (half-life, 1.3 days) exhibited substantial migration through the unsaturated zone, after their application. Two non-persistent herbicides, namely metolachlor (half-life, 21 days), and glyphosate (half-life, 15 days), were not found in the soil or across the vadose zone. Haloxyfop, a moderately persistent (55 days, half-life) pesticide, was reported above the detection limit for soil samples to 50 cm depth just after application in 2019 but was not found throughout the vadose zone. The persistent

imazapic (187 days, half-life) applied in 2014, was not identified in the soil or across the vadose zone during the study period. However, among the detected pesticides, the persistent imidacloprid (half-life, 187 days) was not used at the site since 2013 but found in soil samples and in the vadose water samples (Table 2). At the last sampling period (December 2019), none of seven pesticides were not detected through the vadose zone, indicating their transport beyond the vadose zone or ultimate degradation across the vadose zone.

**3.2.1 Fluroxypyr**

Fluroxypyr was transported beneath the sugarcane root zone, immediately after its application, and the decreasing trend of its concentration over time indicated its ultimate degradation within the vadose zone (Fig. 6). It was also below the detection limit through the soil, from 0-55 cm depth. Previous studies detected small residual loads of the active ingredients of fluroxypyr with 0.03 kg a.i./ha at 30 cm depth of soil (Van Zwieten et al., 2016). Due to its natural decay, moderately

persistent fluroxypyr (half-life, 51 days) was not supposed to be identified in any vadose water sampled in November 2017. Fluroxypyr was applied once at the site in December 2017 (Table 2). Subsequent sampling in January 2018 showed the detection of fluroxypyr at the ports ranging from 1.15 to 2.87 m, but not at the deeper VSPs (Fig. 6). Following volatilization and root uptake, fluroxypyr was transported below the root zone, up to 2.87 m depth from the surface within 24 days after its application. These observations indicate the transport of fluroxypyr to a depth of at least 2.87 m below the soil surface (Fig.



6). These results provide complementary evidence that water is draining vertically downwards below the seasonal water
table (perched layer) and through the entire vadose zone.

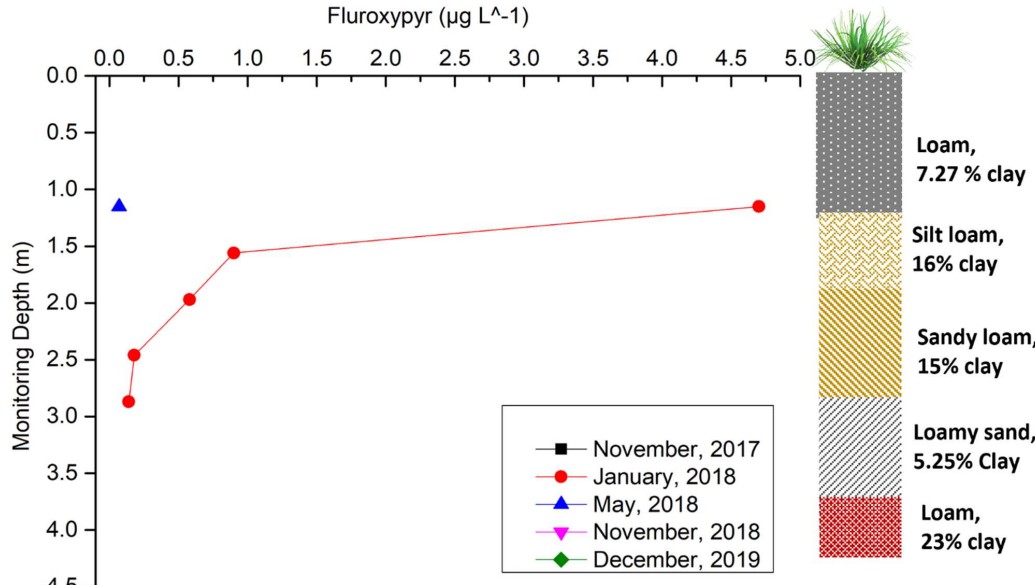

**Figure 6.** Fluroxypyr concentration over time in depth (November 2017, November 2018 and December 2019 were not shown in the figure as the results were below the detection limit).

In January 2018 sampling, fluroxypyr was observed at a concentration of 4.70 µg/L at the upper port of 1.15 m depth, one
month after its application. The concentration levels were exceeding the detection limit (0.10 µg/L), following a sharp
decreasing trend with depth, at the ports of 1.56 m, 1.96 m, 2.46 m, and 2.87 m depth. Beyond 2.87 m from the surface, it
was below the detection limit (Fig. 6). The decreasing trend could be due to the loam rich layers within the soil profile at 0-
55 cm, which could slow down the pesticide transport through the vadose zone and enable transformation of fluroxypyr into
its metabolites (Tao & Yang, 2011). This rapid degradation could cause the absence of fluroxypyr and its metabolites in the
deepest VSPs (3.69 m or 4.00 m in depth) during the monitoring period. In addition, when the soil was saturated, the lateral
flow occurred before vertical leaching, below 1-2 m depth could take place (Masters, et al., 2017). This could be another
potential reason for the sharp decline of concentration below 1.15 m depth.

Fluroxypyr was detected only in the topmost port (1.15 m in depth) in samples collected May 2018. As fluroxypyr and its
metabolites were not found in the vadose water, it is likely that fluroxypyr and its metabolites had completely metabolized
by May 2018. Additionally, from January to April 2018, high water content (> 40%) through the vadose zone (Fig. 4) and
the partial flood event in March 2018 at the monitoring site (Masters et al., 2017) could also contribute to a quicker



degradation of fluroxypyr within this period (Tao and Yang, 2011). Therefore, there is limited potential for the fluroxypyr to reach regional groundwater through direct vertical transport through vadose zone. This is also supported by the lack of
detections of fluroxypyr within the vadose zone water sampled in November 2018, and in December 2019. Residues of fluroxypyr and its metabolites were also no longer detected in the soil profile during November 2018-December 2019.

### 3.2.2 Imidacloprid

Persistent imidacloprid (half-life, 187 days) was found in soil samples and in the vadose water samples, four times, from 1.15 m to 4.00 m depth. Although imidacloprid was not applied in the monitoring site, its transport and concentration
beneath the sugarcane root zone varied with depth over time. It was detected through the vadose zone in November 2017 (Fig. 7). As it was observed in the soil samples, it was possibly being released from residues present in the soil. During September-November 2017, there were several medium to very high rainfall events which resulted in infiltration (Fig 4). This could have contributed to imidacloprid leaching beyond the root zone and travelling to 4.0 m depth, as it has characteristics of high solubility (610 mg/L) and high leachability (GUS Leaching Potential Index, 3.69, Table A1).

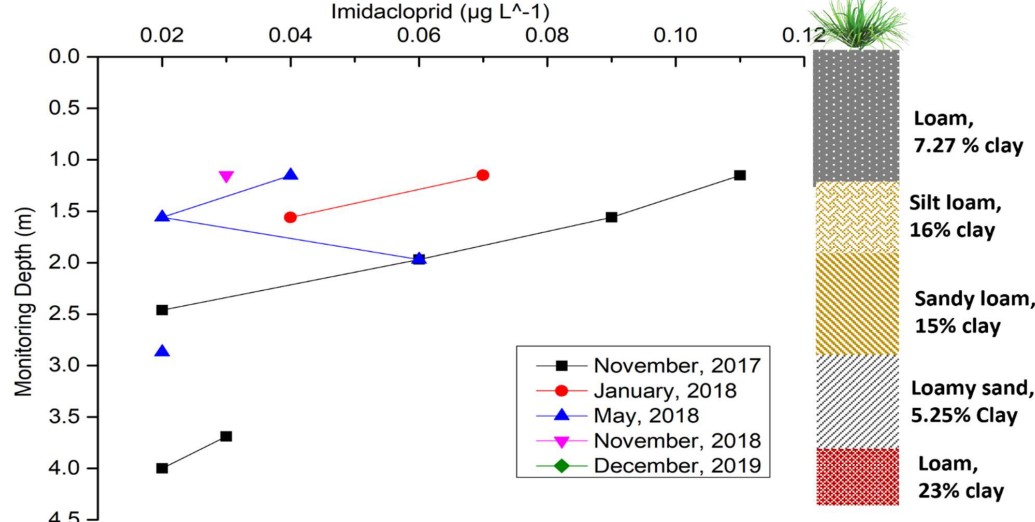


**Figure 7.** Imidacloprid concentration over time in depth (December 2019 was not shown in the figure as the result was below the detection limit).

After November 2017, imidacloprid was detected only in the upper vadose zone and not at the end of the vadose zone (3-4 m depth), in the proximity of the perched aquifer. Imidacloprid was found to a depth of 1.56 m depth in January 2018, to a
depth of 2.96 m in May 2018 and to a depth of 1.15 m in November 2018. There was also a reduction in imidacloprid



concentrations observed over time. This lowering concentration could be due to the combination of the dilution by infiltration and lateral transport after consecutive high rainfall events, as drainage water was reported to seep laterally into the neighbouring channels at the site (Masters et al., 2017). Finally, in December 2019, imidacloprid was below the detection limit throughout the vadose zone, though it was observed in the soil samples. The possible explanation for the lower leaching

could be its sorption onto the clay rich sediments (Oi, 1999) and lower rain events in December 2019 (Fig. 4) (Gupta et al., 2002).

### 3.2.3 Isoxaflutole

Isoxaflutole, applied annually during 2015 to 2018, was not found through the entire depths of vadose zone during the five sampling periods (Fig. 8). As it is quickly transferred into a diketonitrile derivative (DKN) in soils (Mougin et al., 2000), it

can be undetectable immediately after application. The diketonitrile derivative, an active pesticide with a longer half-life and higher water solubility than the parent pesticide, undergoes rapid conversion to the inert benzoic acid analogue (Mitra et al., 2000; Mougin et al., 2000). It has also been reported to bind firmly to soils with higher organic matter content and retained a large portion in soil, resulting in lower leaching potential for DKN (Mitra et al., 2000). Yet, it was not found in the 0-55 cm depth soil profile, despite the high organic material content (5.7%) (Masters et al., 2017). The fact that no isoxaflutole was

identified in the soil profile at this site is not surprising as this pesticide is generally not found below 6 cm of soil depth (Epa, 1998).

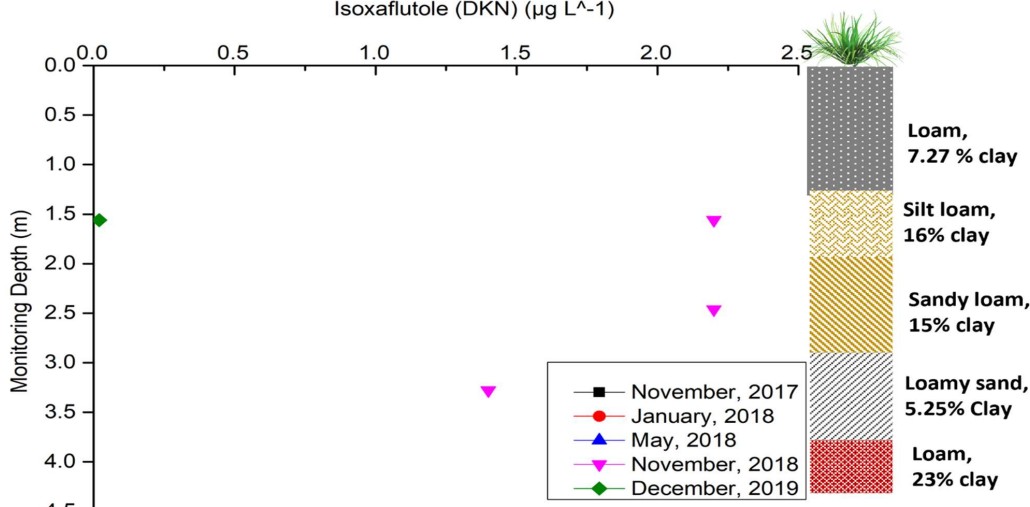

**Figure 8.** Isoxaflutole concentration over time in depth (November 2017, January 2018 and May 2018 were not shown in the figure as the results were below the detection limit).





The concentration of isoxaflutole metabolite (DKN) and its inactive benzoic acid derivative were below the detection limit at all depths sampled in November 2017 before it was applied in December 2017. As isoxaflutole has low half-life (1.3 days), it was not found in samples collected in January 2018 and May 2018. The scenario was different for samples collected on November 2018 (one month after isoxaflutole application) as DKN was found in the middle of the vadose zone (1.56 – 3.28 m depth). The concentration was at 2.2 µg/L at depths of 1.56 m and 2.46 m and 1.4 µg/L at 3.28 m depth. The

concentrations decreased with depth but were higher than the ecotoxicity limit (0.46 µg/L). It is worth noting that isoxaflutole was not detected at the two ports at 1.97 m and 2.87 m depth, despite their proximity. This could be due to the lateral heterogeneity. After harvesting the final ratoon in 2019, isoxaflutole was just above detection limit only at 1.56 m depth, even though it had not been applied. This indicates that isoxaflutole can be transported across the unsaturated zone and may possibly also reach groundwater. Yet, its high degradation rate makes it unlikely that it will be transported into

groundwater.

### 3.2.4 Haloxyfop

Haloxyfop, a moderately persistent (55 days, half-life) pesticide, was found to be below the detection limit for soil samples in 2018 but it was reported above the detection limit just after application in 2019 (Table 3). Haloxyfop has the GUS leaching potential index, 3.70 (high leachability), indicating higher likelihood of transport through the soil (PPDB, 2021;

Gustafson, 1989). The concentration of haloxyfop at different depths of 25 cm, 35 cm and 50 cm were 0.012 kg a.i./ha (0.004 mg/Kg), 0.043 kg a.i./ha (0.01 mg/Kg), and 0.012 kg a.i./ha (0.002 mg/Kg), respectively. These results indicate the transport of haloxyfop to at least 50 cm soil depth. They also indicate the potential for haloxyfop to travel beyond this depth, through the vadose zone. But it was below the detection limit throughout the vadose zone. This indicates that it did not reach 1.15 m depth in short time (two weeks). The transport of haloxyfop could be reduced due to lack of infiltrating rainfall and

subsequent low water content across the vadose zone in this period.

### 3.4 Pesticide concentration in groundwater

Only two pesticides were found in three of the six bores, sampled once, in the South Johnstone River sub-basin in July 2017. Imidacloprid was exposed in two bores RN11210032 and RN11210051, at concentrations with 0.60 µg/L and 0.08 µg/L, respectively. Compared to the recently developed aquatic ecosystem protection guideline values for the GBR (King et al.,

2017b, a; Smith, 2018), the concentration of imidacloprid in bore RN11210031 exceeded the freshwater ecosystem health guideline (0.11 µg/L) for protection of 95% of species (Table 5). Previous studies also detected imidacloprid at 0.06 µg/L in the South Johnstone River sub-basin surface waters (Smith et al., 2012) and 1.15 µg/L in bores with in the Johnstone catchment (Masters et al., 2014). Though imazapic was not detected previously in any surface water and groundwater in the South Johnstone catchment (Masters et al., 2014; Smith et al., 2012), it was detected only in bore RN11210056. Its

concentration, 0.70 µg/L, was greater than the ecotoxicity value, 0.41 µg/L (Table 5). Other herbicides, namely metolachlor, fluroxypyr, isoxaflutole, glyphosate and haloxyfop, were all below the detection limits in the six groundwater monitoring





bores tested. None of these pesticides have previously been found in surface water studies in the basin (Smith et al., 2012) or in bores with in the Johnstone catchment (Masters et al., 2014). The lack of detections of these pesticides in surface and groundwater most likely relates to their limited use in recent decades combined with natural degradation of these pesticides.

Overall, the pesticide results of the two site monitoring bores (RN183021 and RN183022, collected five times, at the same time as vadose water sampled through VSP) were below the detection limits for all seven pesticides tested in this study. Significantly, isoxaflutole, applied annually, was expected to be present in the soil and in the vadose zone and potentially leached to groundwater. It has previously been reported that the annual pesticide mass leached below the root zone could range between <0.1 and 1% (sometimes higher, up to 4%) (Flury, 1996). The lack of isoxaflutole and its metabolites in the

soil sample and declining concentration in the pore water with depth and time highlight limited transport through the vadose zone. This could be due to the low application dose of 150 g/ha isoxaflutole in the field trial. Additionally, the amount of isoxaflutole percolated below the root zone after plant consumption could be degraded into the soil before reaching to the groundwater. This is supported by its absence in the monitoring bores at the site.

**4 General Discussion**

The infiltration process is exemplified by the soil moisture content in the shallow soil profile, which exhibits a seasonal perched water table at 1.56 m and 1.97 m depth (Fig. 4). Under this perched layer, unsaturated conditions occurred at depths of 2.46 m and 2.87 m. Frequent monitoring of regularly applied pesticides supported interpretation of the hydrological characteristics through the vadose zone. This was evidenced by the concentration levels of fluroxypyr exceeding the detection limit (0.10 µg/L) and following a sharp decreasing trend with depth, from 1.15 m to 2.87 (Fig. 6). The decreasing

trend could be due to the combination of reduced fluroxypyr transport by the clay layer, transformation of fluroxypyr into its metabolites by carbon content (Tao & Yang, 2011), drainage by lateral flow up to 1-2 m depth at the site (Masters, et al., 2017) or dilution because of the high-water content in the perched layer.

Fluroxypyr was not identified in the deepest part of the vadose zone (3.69 m or 4.00 m in depth) where a perched aquifer layer exists. This observation supports the regulating and partitioning behaviour of the red coarse mottled structured loam

layer (with 23% clay) at 3.80-4.00 m depth. This layer could control the ability of rainwater infiltration as well as pesticides to reach the more permeable aquifer material at ~9 m (ground water). However, the results of non-PS II herbicides were below the detection limit in groundwater, indicating less potential to be leached to groundwater.

Based on the PIRI (Pesticide Impact Rating Index), imidacloprid and isoxaflutole were predicted to present 'low' risk profiles and metolachlor 'high' risk to invertebrates (Davis et al., 2014). In this study, imazapic and metolachlor, last applied

in 2014 at the site, were not identified in the soil profile or across the vadose zone and imidacloprid was below the detection limit through the vadose zone at the end of sampling. As this study found that those non-PS II herbicides (imidacloprid, isoxaflutole and metolachlor) were attenuated within the soil profile within in study period (about three years), this study substantiates the switch of PS II herbicides to non-PS II herbicides. Our data also supports the substitution for other





herbicides (fluroxypyr and haloxyfop, which were projected to present 'medium' risk profiles) (Davis et al., 2014). The
findings of glyphosate used at the study site in 2013-2014 and 2019 also supporting its substitution, yet it was predicted
'high' risk to invertebrates (Davis et al., 2014).

This study, focussing on seven non-PS II herbicides in the Wet Tropics, has provided increased knowledge of the fate,
existence, and transport of these pesticides across the vadose zone. As the concentration of regularly applied pesticides in the
vadose zone reduces quickly following pesticide application, a shift of product choice to alternative/other non-PS II
herbicides instead of PS II herbicides could also be advisable in other tropical regions with agricultural activities.

This study describes improved characterisation of pesticide movement dynamics (after immediate application) under a
sugarcane field. This allows a better understanding of agrochemical impact from sugarcane farming on the groundwater
environment in the tropics. Only two pesticides (imidacloprid and imazapic) were detected in three of the six bores sampled
once in the South Johnstone River sub-basin, but these pesticides were not detected at the bores on the monitoring site. As
there was no record of the detections of these two pesticides at the end of the monitoring period, these were ultimately
degraded in the groundwater to concentrations below current detection limits. However, based on the data at one site, the
study was not able to explore the potential contamination at a regional groundwater scale.

## 5 Conclusions

Rainwater percolation through the soil and unsaturated zone resulted in pesticide transport to deeper components of the
vadose zone and into groundwater, but the extent of their transport beyond the root zone of sugarcane was attributed to the
rainfall events following pesticide application. Fluroxypyr and haloxyfop are both moderately persistent with high
leachability (GUI, >2.80). Fluroxypyr application followed by several rainfall events, showed substantial migration to a
depth at least 2.87 m below the soil surface but haloxyfop did not, as the lack of infiltrating rainfall and subsequent low
water content through the vadose zone may have reduced its transport beyond the root zone. However, two non-persistent
herbicides, namely isoxaflutole and glyphosate (with low leachability, GUI, <1.80), are unlikely to reach the vadose zone.

The persistent imazapic, applied in 2014, was not expected to be detected in the soil or across the vadose zone during the
study period. However, among the detected herbicides, the persistent imidacloprid was not utilized in the site since 2013 but
found in soil and in the vadose water. None of seven pesticides were identified within the vadose zone at the last sampling
period (December 2019). Based on the data, this present study revealed the transport of the non-PS II pesticides beyond the
crop root zone, but these pesticides were no longer detectable during the last sampling period. Their disappearance is due to
either transformation into its metabolites by the thick carbon rich clay layer within the soil profile or dilution within the
perched layer, thereby indicating their limited ability to reach groundwater. The findings of this case study support the
substitution of PS II herbicides with the non-PS II pesticides (at least seven of all other pesticides). However, some non-PS II
pesticides, for example, persistent imidacloprid and imazapic, were found at concentrations higher than the ecotoxicity



threshold level in the regional aquifer groundwater samples, indicating that at some locations and with some products, contamination of the groundwater is occurring.

**Data availability.** To access the raw data, contact with the corresponding author.

**Author contributions.** Dr Lucy Reading formulated the research questions and directed the research. Professor Ofer Dahan assembled and inaugurated the vadose zone monitoring system at the monitoring site. Dr Rezaul Karim collected and
analysed the data extensively and prepared the draft of the manuscript. Prof. Les Dawes, Prof. Ofer Dahan and Dr Glynis Orr reviewed and edited. Dr Karim finalized the manuscript for submission and production.

**Declaration of competing interests.** The authors have no known declaration of competing interests that could have appeared to influence the work reported in this paper.

**Acknowledgments.** The authors would like to convey their gratitude to Jim Stanley and Jim Yaxley, Queensland University
of Technology and Star Drilling, Thomastown for assembling and inaugurated the monitoring system at the monitoring site. We thank Christian Brunk from the Queensland University of Technology and Daniel Roebuck from the Campbell Scientific Australia for their technical assistance in functioning the experimental setups. We thank the Centre for Wet Tropics Agriculture, South Johnstone. Further thanks are due to the dedicated Paddock-to-Reef project, Silkwood team for their concerted supports with sharing information, project outputs and assisting with field work. We thank to MSF Sugar Pty Ltd
for giving entrance to their land and crop-pest management records. Gratitude is also expressed for the extensive laboratory analysis provided by Queensland Health, Coopers Plains.

**Financial supports.** The funding for this project was provided by the Department of Regional Development, Manufacturing and Water, Australia, and the Queensland University of Technology.

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



**Appendix**

**Table A1.** Pesticides applied at the block and their properties.

| Pesticide | Chemical formula[a] | Used for[b] | Mode of action[b] | Solubility (mg/L at 20°C)[a] | Adsorption (Koc); Mobility[a] | Half-life (DT$_{50}$); Persistence[a] | GUS Leaching Potential Index[a*] | Degradation mechanism[b] |
|---|---|---|---|---|---|---|---|---|
| Imidacloprid | 1-(6-chloro-3-pyridylmethyl)-N-nitroimidazolidin-2-ylideneamine | Sucking insect | Irreversible acetylcholine receptors blockage | 610; High | moderately mobile | 187; Persistent | 3.69; High leachability | Rapidly broken down in water by sunlight |
| Imazapic | 2-5methyl-3-pyridinecarboxylic acid | Pre- and post-emergent grasses | AHAS enzyme, blocking protein synthesis and cell growth inhibitor | 2230; High | 137; Moderately mobile | 232; Persistent | 4.41; High leachability | Primarily by microbial metabolism; Does not volatilize |
| Metolachlor | 2-chloro-60-ethyl-N-(2-methoxy-1-methylethyl) aceto-o-toluidide | Broadleaf and annual grassy weeds | Gibberellic acid biosynthesis inhibitor | 480; Moderate | 200[c] Moderately mobile | 21; Non-persistent | 3.29; High leachability | Biological degradation; Moderately adsorbed by most soils |
| Fluroxypyr | 4-amino-3,5-dichloro-6-fluoro-2-pyridyloxyacetic acid | Broadleaf weeds | Auxin growth regulator | 6500; High | 5[c], Very mobile | 51; Moderately persistent | 3.70; High leachability | Primarily by hydrolysis; Microbial metabolism |
| Isoxaflutole | 5-cyclopropyl isoxazol-4-yl-2-mesyl-4-trifluoromethyl phenyl ketone | Pre-emergence herbicide for grass and broad leaf weed | Carotenoid pigments inhibitor | 6.2; Low | 145; Moderately mobile | 1.3; Non-persistent | 0.24; Low leachability | Rapid degradation under field conditions |
| Glyphosate | N-(phosphonomethyl) glycine | Annual and perennial plants | Shikimic acid inhibitor | 10500; High | 1424; Slightly Mobile | 15; Non-persistent | -0.29; Low leachability | Primarily slow microbial metabolism; Strongly adsorbed to soil |
| Haloxyfop | 2-(4-((3-chloro-5-(trifluoromethyl)-2-pyridinyl)oxy) phenoxy) propanoic acid | Grass weeds | Acetyl CoA carboxylase inhibitor | 1.6; Low | 75; Moderately mobile | 55; Moderately persistent | 3.70; High leachability | Biological degradation in soil |

[a] (PPDB, 2021), [b] Tu et al. (2001); [c] Senseman (2007); * GUS values lower than 1.8 and higher than 2.8 indicate, respectively, non-leacher and leacher pesticide compounds; for GUS values between 1.8 and 2.8 the pesticide is considered 650 in a transition zone (Gustafson, 1989).