# Peer review of "Pesticide transport through the vadose zone under sugarcane in the Wet Tropics, Australia"

_EGUsphere, 2022_

## Author Response (AR1)

***The reviewers' comment are addressed in the revised manuscript. The issues (comments) with incorporated in the text are provided briefly below.***

**The analytical procedures (calibration, LOD, LOQ, recovery, matrix effect,… should be included in section 2.5. Chemical analysis.**

Two subsections (2.5.1 Water analysis and 2.5.2 Soil Analysis) are added under 2.5 Chemical analysis (Below). The analytical methods and sample processing used in Queensland Health Forensic and Scientific Services are added. For the analytical procedures, the LOR and recovery are added in the text and subsequently in the Appendix Table A2 and Table A3 (attachment).

**"2.5 Chemical analysis**

The pesticides and their possible metabolites were analysed by Queensland Health, Coopers Plains, Queensland. All collected samples (water and soil) were separated via solid phase extraction (SPE) and examined by liquid chromatography mass spectrometry (LCMS). The analytical methodology (SPE combined with LCMS) is generally utilised by the Great Barrier Reef Catchment Loads Modelling Program (Gallen et al., 2016; Shaw et al., 2010; Vardy et al., 2015; Wallace et al., 2015). One insecticide (imidacloprid) and six herbicides namely imazapic, metolachlor, fluroxypyr, isoxaflutole, glyphosate and haloxyfop, which have been applied in recent years at the monitoring site, were tested in this study (Table 5 and Table A1).

2.5.1 Water Analysis

There were two analytical method groups (QIS 33963 for herbicides and pesticides and QIS 33917 for Glyphosate) for water analysis used in Queensland Health Laboratory. The analysis was performed by direct injection method by filtering 1mL of sample using 0.2 µm filter and analysed on LCMSMS. For water herbicide analysis for imidacloprid, imazapic, metolachlor, fluroxypyr, isoxaflutole and haloxyfop, the method details are provided in Table A2. During analysis, some matrix effects were experienced, and if this increased, the Limit of Reporting (LOR) was increased.

2.5.2 Soil Analysis

There were two analytical method groups (QIS 30814 for glyphosate and QIS 32456 for herbicides and pesticides) in soil / sediment. For QIS 30814: glyphosate and amino methyl phosphonic acid (AMPA) in soil/vegetation by LCMSMS, water was added to soil samples and shaken. The aqueous phase was filtered and analysed via direct injection on the LC-MSMS. On the other hand, QIS 32456: determination of herbicides in soil and sediment by LC-HRAM-Orbitrap, the soil/sediment sample was first shaken with acetone using a tabletop shaker for approximately 12 hours. The herbicides were then extracted using a QuEChERS method. The final extract was analysed by LC-HRAM-Orbitrap. The method details for pesticide and herbicides in soil and sediment are provided in Table A3. Imazapic showed low recovery (<40%) when it was analysed by QIS 33456 method."

**Table A2.** Method details for Pesticides in Water by Direct Injection using LCMSMS and QExactive Orbitrap.

| Pesticide | LOR | Units | Accepted (i.e., expanded) Uncertainty (%) | Recovery (%) | Repeatability (r) (%) | Standard Uncertainty (%) |
|---|---|---|---|---|---|---|
| Fluroxypyr | 0.05 | ug/L | 28 | 111 | 38 | 28 |
| Haloxyfop (acid) | 0.02 | ug/L | 26 | 105 | 31 | 26 |
| Hexazinone | 0.01 | ug/L | 25 | 103 | 20 | 12 |
| Imazapic | 0.01 | ug/L | 25 | 101 | 14 | 10 |
| Imidacloprid | 0.02 | ug/L | 25 | 100 | 34 | 21 |
| Imidacloprid (metabolites) | 0.02 | ug/L | 34 | 108 | 21 | 34 |
| Total Imidacloprid | 0.04 | ug/L | 25 | | | |

| Pesticide | LOR | Units | | Recovery % | | |
|---|---|---|---|---|---|---|
| Isoxaflutole metabolite (Diketonitrile) | 0.02 | ug/L | 25 | 102 | 20 | 16 |
| Metolachlor | 0.01 | ug/L | 25 | 87 | 16 | 11 |

**Table A3.** Method details for Pesticides in Soil and Sediment by LCMSMS / LC-HRAM-Orbitrap.

| Pesticide | Limit of reporting | Units | Recovery % | |
|---|---|---|---|---|
| Fluroxypyr | 0.001 | mg/kg | 55 | |
| Haloxyfop (acid) | 0.001 | mg/kg | 69 | |
| Imazapic | 0.001 | mg/kg | 22 | |

**The reviewers' comment (2) is addressed in the revised manuscript. The issues (comments) with incorporated in the text are provided briefly below.**

*1. Include the name of abbreviations in all Tables and Figures used as Table foot or in the Figure caption to facilitate understanding.*

We have added the abbreviated in the required Tables and Figures i.e., Figure 3, 4.

*2. L111. Can you please give the soil type classification according to a more international system to be understood by most readers? Such as Key to Soil Taxonomy of USDA or World Reference Base for Soil Resources of IUSS/FAO.*

Regarding soil classification, the following sentences are added into section 2.1.

"Based on the world reference base for soil resources 2014 (IUSS Working Group WRB, 2022), the site soil is classified as 'Stagnic Umbrisol (Pantoclayic, Sideralic, Humic)' by Enderlin and Harms, 2023.

*3. Also include pH values besides organic matter, important to understand biochemical processes in soil.*

pH is an important parameter to understand the biochemical processes in soil, however the soil pH was not measured during the analysis period.

*4. L123-129. Can you also add mean annual T and potential evapotranspiration?*

Regarding annual T and potential evapotranspiration, the following sentences are added into section 2.1.

"According to the interpolated climatic data available through the Queensland Government's SILO database from 2010 to 2023 (at SILO Grid point: Latitude – 17.75 and longitude - 146.05), the mean annual minimum and maximum temperatures are 21ºC and 29ºC respectively. The average predicted potential evapotranspiration is 5.18 mm/yr, determined by Morton's potential Evapotranspiration, during 2010-2023. The annual average interpolated rainfall is 3,202 mm/yr at the monitoring site. According to Bureau of Meteorology (BoM), the annual average rainfall is 3,092 mm/yr at the nearest BoM Station–Bingil Bay (Site 032009), situated nearly 12.5 km of the site".

*5. Table 3. Please, explain what you mean with X and √ and what the difference is. Not clear. Table 4. Please explain what you mean with "√" and "sampling regime"*

The table captions have been updated:

Water sampling regime for pesticide analysis (√ denotes samples collected on the corresponding date and X denotes **dates when no samples were collected**).

*6. Figure 4. Explain in the figure 4 what A, B and C represent with the circles*

In the figure 4, A, B and C were used to refer some to some specific situations of water dynamics, which are discussed in the text.

*7. L326. It is written "As it was observed in the soil samples,...", but where are the soil results? Not clear. Difficult to assess what is soil and what is water. I do not know where to find them. I only see results, but not explicitly indicated along the text, on Supplementary Tables S8 and S9. Clear information about results in soils should be provided.*

Following the results of Table S2-9, we rewrote the section 3.2.2 Imidacloprid, with adding relevant texts and referring the tables.

**"3.2.2 Imidacloprid**

During the monitoring time 2017-2019, persistent imidacloprid (half-life, 187 days) was found in the vadose water samples (Fig. 7), four times (Table S 2, 3, 4 & 5), from 1.15 m to 4.00 m depth. Although imidacloprid was not applied in the monitoring site, its transport and concentration beneath the sugarcane root zone varied with depth over time. It was possibly being released from residues present in the upper soil layers or transported imidacloprid from the neighbouring sugarcane fields. During September-November 2017, there were several medium to very high rainfall events which resulted in infiltration (Fig 4). This could have contributed to imidacloprid leaching beyond the root zone and travelling to 4.0 m depth, as it has characteristics of high solubility (610 mg/L) and high leachability (GUS Leaching Potential Index, 3.69, Table A1).

After November 2017, imidacloprid was detected only in the upper vadose zone and not at the end of the vadose zone (3-4 m depth), in the proximity of the perched aquifer. Imidacloprid was found to a depth of 1.56 m depth in January 2018, to a depth of 2.96 m in May 2018 and to a depth of 1.15 m in November 2018 (Fig. 7). There was also a reduction in imidacloprid concentrations observed over time. This lowering concentration could be due to the combination of the dilution by infiltration and lateral transport after consecutive high rainfall events, as drainage water was reported to seep laterally into the neighbouring channels at the site (Masters et al., 2017). Finally, in December 2019, imidacloprid was below the detection limit throughout the vadose zone (Fig. 7). The possible explanation for the lower leaching could be its sorption onto the clay rich sediments (Oi, 1999) and lower rain events in December 2019 (Fig. 4) (Gupta et al., 2002).

Imidacloprid was not detected in soil samples collected in September and November 2018 (Table S8). Interestingly, persistent imidacloprid was found at upper part of the soil at 25 cm and 35 cm depth with 0.0122 kg a.i./ha (0.004 mg/Kg) and 0.010 kg a.i./ha (0.01 mg/Kg), and 0.012 kg a.i./ha (0.002 mg/Kg), respectively, in November 2019 (Table S9). It was observed only at 25 cm soil depth, with 0.0366 kg a.i./ha (0.012 mg/Kg) in December 2019. The concentration of imidacloprid at upper part of soil may indicate a source within a neighbouring sugarcane field. This imidacloprid had not yet travelled into the vadose zone till 1.15 m depth or beyond as it was not detected in vadose water sampled in November and December 2019 (Fig.7)."

*8. In addition, concentration in Tables S8 and S9 is in ug/L, what it is a unit for liquids. Give a concentration for the soil as ug/kg for example. Concentration of pesticides on soil should be in the manuscript, clearly described and referred.*

There was a typo error for the unit of Pesticide in soils. It should be mg/Kg. This has now been corrected in the Tables S8 and S9. The reference Table for S8 and S9 are inserted into the relevant text.

**Table S8.: the pesticide and their residues in soil (September and November 2018)**

| Herbicides in water | Health value | Units | Reporting Limit | September, 2018 | | | November, 2018 | | |
|---|---|---|---|---|---|---|---|---|---|
| | | | | 25 cm | 35 cm | 50 cm | 25 cm | 35 cm | 50 cm |
| Glyphosate | 1000 | mg/Kg | 0.70 | < | < | < | < | < | < |
| Fluroxypyr | | mg/Kg | 0.01 | < | < | < | < | < | < |
| Hexazinone | 400 | mg/Kg | 0.01 | < | < | < | < | < | < |
| Imazapic | | mg/Kg | 0.01 | < | < | < | < | < | < |
| Imadacloprid | | mg/Kg | 0.02 | < | < | < | < | < | < |
| Isoxaflutole metabolite (DKN) | | mg/Kg | 0.02 | < | < | < | < | < | < |
| Metolachlor | 300 | mg/Kg | 0.005 | < | < | < | < | < | < |
| Haloxyfop | | mg/Kg | 0.001 | < | < | < | < | < | < |

**Table S9.: the pesticide and their residues soil (November and December 2019)**

| Herbicides in water | Health value | Units | Reporting Limit | November 2019 | | | December 2019 | | |
|---|---|---|---|---|---|---|---|---|---|
| | | | | 25 cm | 35 cm | 50 cm | 25 cm | 35 cm | 50 cm |
| Glyphosate | 1000 | mg/Kg | 0.70 | < | < | < | < | < | < |
| Fluroxypyr | | mg/Kg | 0.01 | < | < | < | < | < | < |
| Hexazinone | 400 | mg/Kg | 0.01 | < | < | < | < | < | < |
| Imazapic | | mg/Kg | 0.01 | < | < | < | < | < | < |
| Imadacloprid | | mg/Kg | 0.02 | 0.004 | 0.010 | < | 0.012 | < | < |
| Isoxaflutole metabolite (DKN) | | mg/Kg | 0.02 | < | < | < | < | < | < |
| Metolachlor | 300 | mg/Kg | 0.005 | < | < | < | < | < | < |
| Haloxyfop | | mg/Kg | 0.001 | < | < | < | 0.004 | 0.010 | 0.002 |